

Reefal ostracod assemblages from the Zanzibar Archipelago (Tanzania)
Skye Yunshu Tian[1], Martin Langer[1], Moriaki Yasuhara[2,3], Chih-Lin Wei[4]
[1]Steinmann-Institut für Geologie, Mineralogie und Paläontologie, Universität Bonn,
Bonn, Germany
[2] School of Biological Sciences, Area of Ecology and Biodiversity, Swire Institute of
Marine Science, The University of Hong Kong, Hong Kong SAR, China
[3] State Key Laboratory of Marine Pollution, City University of Hong Kong, Hong Kong
SAR, China
[4] Institute of Oceanography, National Taiwan University, Taipei 106, Taiwan
Correspondence to: Skye Yunshu Tian skyeystian@gmail.com
Martin Langer martin.langer@uni-bonn.de
**Abstract**
Tropical reefs encompass tremendous biodiversity yet are imperiled by increasing
natural and anthropogenic disturbances worldwide. Meiobenthic biotas on coral reefs,
for example, ostracods, may experience substantial diversity loss and compositional
changes even before being examined. In this study, we investigated the reefal ostracod
assemblages from the highly diverse and productive ecosystem in Zanzibar
Archipelago (Pemba, Zanzibar, and Mafia islands), Tanzania, to understand how their
diversity and faunal structure vary in response to water depth, benthic community type,
and human impacts. We characterized four distinct ostracod faunas associated with
different benthic habitats, which were deep fore reefs, shallow fringing reefs, degraded
fringing reefs, and algal covered intertidal flats. We identified typical ostracod
associations, i.e., Bairdiidae versus Loxoconchidae-Xestoleberididae, that showed
affinities to hard corals or algae on the reef platforms, respectively. Highest diversity
was found on shallow fringing reefs where reefal and algal taxa exhibited maximum
overlap of their distributional ranges, while the sand flats, mangrove, and marginal reefs
within the intertidal zone had much lower diversity with high dominance of euryhaline
taxa. Along the western coast of Zanzibar, coastal development likely resulted in a
unique faunal composition and comparatively low diversity of ostracod assemblages
among those in reefal habitats, in conjunction with overall reef ecosystem degradation.
This study represents the first large-scale assessment of shallow-marine ostracods in



the Zanzibar Archipelago. It lays a solid foundation for future research into the
ecological significance of ostracods on coral reefs.
**1 Introduction**
Coral reefs as the most diverse ecosystem in the marine realm hold great ecological and
economic values, yet our knowledge of its enormous biodiversity is far from complete.
Compared with well-studied, conspicuous macrofauna (Souza et al., 2023), meiofauna
on coral reefs are highly under-represented in current research despite being
ecologically essential components and contributing significantly to total biodiversity
(Leray and Knowlton, 2015; Plaisance et al., 2011). Ostracoda (Crustacea) among all
meio-benthos has a tight association with reef environments tracing back to the lower
Paleozoic (Whatley and Watson, 1988). As a useful model organism in modern and
paleo biodiversity research because of its high fossilization potential, high abundance,
and ubiquity in almost all marine ecosystems (Yasuhara et al., 2017), does ostracod
exhibit higher diversity in reefal habitats compared with other soft sediment
environments? What are the characteristic ostracod taxa occupying different niches on
coral reefs? Answers to these questions are important for a holistic understanding of
the reef ecosystem and may hint at the underlying mechanisms that support such
extraordinary reef diversity. With intensifying anthropogenic disturbances at local to
global scales, the need to examine reefal ostracods before they perish is pressing.
Studies targeting tropical shallow-marine ostracods on coral reefs are surprisingly
deficient. Across the circumtropical belt, the central Indo-Pacific receives the most
attention for its diverse reefal ostracods, with pioneering studies identifying distinct
faunas associated with depth habitats from the shallow intertidal to deep reefal zones
(Whatley and Watson, 1988; Babinot and Degaugue-Michalski, 1996). Apart from
bathymetry, the distribution of reefal ostracods seems also related to benthic
community type (coral reefs versus seagrass/algal beds), sediment type (i.e., sandy
versus muddy deposits), in addition to local hydrology (i.e., exposure to wave energy)
(Weissleader et al., 1989; Whatley and Watson, 1988; Babinot and Degaugue-
Michalski, 1996; Tabuki, 1990, 1987). However, most of these works are confined to
small geographic areas and based on limited (sub)fossil materials. An extensive
regional-scale survey of reefal ostracods has never been conducted. More importantly,
the focus of previous studies mainly revolved around taxonomy, and biogeography to




a lesser degree, while quantitative assessments of biodiversity are largely lacking
(Tabuki, 1987, 1990; Mostafawi et al., 2005). The highest species richness (S=74) was
reported for a reef slope environment in Pulau Seribu, Java (Whatley and Watson, 1988)
in contrast to much lower values at lagoons (S=27-42) (Babinot and Degaugue-
Michalski, 1996; Weissleader et al., 1989) and reef flat (S=34) (Mostafawi et al., 2005).
Reefal ostracods are even less known in other tropical regions outside of the central
Indo-Pacific. Along the eastern coast of Africa, where the reef ecosystem is productive
and biodiverse, the only studies on ostracod assemblages are perhaps Hartmann (1974)
and Jellinek (1993) that document more than 200 species inhabiting the algae facies
and reefal facies across the littoral zone in Kenya. Here we present the first large-scale
study on reefal ostracods from the Zanzibar Archipelago, Tanzania, a biodiversity
hotspot of great conservation interests and vulnerability to increasing anthropogenic
impacts (Grimsditch et al., 2009). We investigated the geographical structure of
ostracod diversity and composition in relation to environmental habitats among three
major islands of Pemba, Zanzibar, and Mafia. We compared the patterns with those of
benthic foraminifera (Thissen and Langer, 2017) to explore complex environmental
controls on the two groups of meio-benthos. This study is a major step towards better
understanding of tropical shallow-marine ostracods in eastern Africa and provides
valuable insight into the ostracod-reef association in general.
**2 Regional setting**
The Zanzibar Archipelago is located along the eastern coast of Tanzania in the Western
Indian Ocean (Fig. 1) (Thissen and Langer, 2017). It belongs to the eastern African
biogeographic province that stretches from Somalia to the northeastern coast of South
Africa (Costello et al., 2017; Obura, 2012). The archipelago is strongly influenced by
the warm, westward-flowing South Equatorial Current and the northward-flowing East
African Coastal Current (Narayan et al., 2022). The western coastlines are more
protected, with generally higher coral coverage, whereas the eastern coastlines are
exposed to large physical disturbances and strong wave energy (Thissen and Langer,
2017). Tides there are semi-diurnal, with a maximum range of 4.5 m and a neap tidal
range of 0.9 m (Thissen and Langer, 2017; Narayan et al., 2022). The islands possess a
great variety of benthic habitats from the littoral to open-water zone, with mangroves,
vegetated sand flats, and reef complexes. Reefs are mainly fringing reefs that are





situated on the narrow continental shelf (Mafia, Zanzibar) or are separated from the
African mainland by the deep Pemba channel (Pemba) (Thissen and Langer, 2017).
Noticeably, the major islands are subject to very different degrees of human exploration,
as Zanzibar is densely populated and highly urbanized while Mafia and Pemba are
largely uninhabited (Narayan et al., 2022). Stone Town and Bawe, in particular, are
faced with a direct discharge of untreated domestic sewage along the western coast of
Zanzibar Island, where moderate levels of reef deterioration have been found with
diversity decrease and coral cover loss (Bravo et al., 2021; Larsen et al., 2023).
Although extensive long-term monitoring is still lacking, previous studies indicate that
Pemba reefs are likely in pristine conditions with the highest coverage of live hard
corals, while Zanzibar reefs are often dominated by dead corals intermingled with algae
and seagrass habitats (Ussi et al., 2019; Larsen et al., 2023; Grimsditch et al., 2009).
No quantitative assessment of reef health has been conducted in Mafia Island,
unfortunately, but our field observations suggested moderate to good conditions at our
sampling sites.



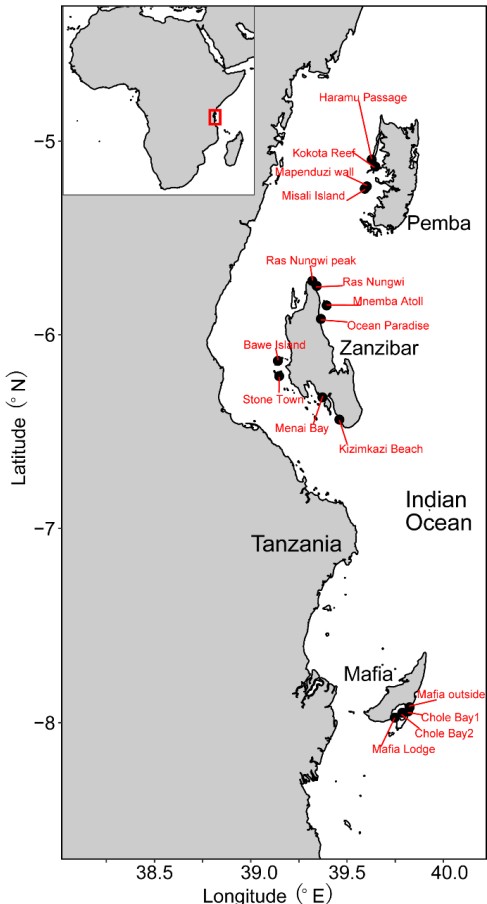

Fig. 1. Locality map showing three major islands of the Zanzibar Archipelago with sample sites.

**3 Materials and methods**

3.1 Samples

26 surface sediment samples were collected from 16 sites during two field campaigns in 2005 at the islands of Zanzibar and Pemba, and in 2012 at Mafia Island (Table 1). Depositional depths of all samples range from 0 to 42 m across the intertidal and subtidal zones. The selected sampling sites cover all major types of benthic habitats, including nearshore mangroves, coastal sand flats, and fringing-, fore-, and back-reefs. Samples were collected by SCUBA diving to fill plastic bags with surface sediments from the top 2 cm.



Most sampling sites were fine to medium-grained carbonate-rich, bioclastic sands and
deposits with some reef rubble. Sediments were washed through a 63 µm sieve and
oven dried at 50 ℃. The residue was dry sieved over a 150 µm mesh sieve and ostracods
were picked from the >150 µm size fraction, because smaller individuals are usually
early juveniles that are not preserved and/or difficult to identify (Yasuhara et al., 2017).
Sediment-rich samples were split into aliquot fractions using a microsplitter. The
sample materials were primarily death assemblages though a small number of
specimens were preserved with soft parts, indicating they were alive at the time of
collection. Both live and dead specimens were included in the total count to represent
time-averaged assemblages, which method effectively defines reef habitats and
provides general environmental and diversity data useful in paleoecology (Glenn-
Sullivan and Evans, 2001; Langer and Lipps, 2003). A single valve or a carapace was
considered as one individual, which is a standard counting method in ostracod research
(Yasuhara et al., 2017). Selected specimens were imaged using a Scanning Electron
Microscope (SEM).
Table 1. Sample information including the geographical position, water depth, habitat
type, in addition to the abundance and raw species richness of ostracod assemblages.

| Location | Island | Depth (m) | Longitude | Latitude | Species richness | Abundance | Habitat |
|---|---|---|---|---|---|---|---|
| Haramu Passage | Pemba | 20 | 39.6280 | -5.0946 | 37 | 69 | fore reef |
| Haramu Passage | Pemba | 30 | 39.6280 | -5.0946 | 35 | 60 | fore reef |
| Kokota Reef | Pemba | 25 | 39.6472 | -5.1311 | 64 | 235 | fringing reef |
| Kokota Reef | Pemba | 16 | 39.6472 | -5.1311 | 78 | 364 | fringing reef |
| Mapenduzi wall | Pemba | 40 | 39.6026 | -5.2334 | 60 | 235 | fore reef |
| Mapenduzi wall | Pemba | 42 | 39.6026 | -5.2334 | 55 | 188 | fore reef |
| Misali Island | Pemba | 20 | 39.5918 | -5.2456 | 65 | 254 | fore reef |
| Ras Nungwi peak | Zanzibar | 12 | 39.3192 | -5.7225 | 56 | 296 | fringing reef |
| Ras Nungwi peak | Zanzibar | 12-14 | 39.3192 | -5.7225 | 46 | 116 | fringing reef |
| Ras Nungwi peak | Zanzibar | 20 | 39.3192 | -5.7225 | 81 | 311 | fringing reef |



| Ras Nungwi | Zanzibar | 16 | 39.3425 | -5.7481 | 92 | 408 | fringing reef |
|---|---|---|---|---|---|---|---|
| Ras Nungwi | Zanzibar | 20 | 39.3425 | -5.7481 | 37 | 76 | fringing reef |
| Mnemba Atoll | Zanzibar | 30 | 39.3939 | -5.8489 | 33 | 87 | sand flat |
| Ocean Paradise | Zanzibar | 3 | 39.3642 | -5.9183 | 46 | 231 | back reef |
| Bawe Island | Zanzibar | 9-30 | 39.1408 | -6.135 | 80 | 410 | fringing reef |
| Bawe Island | Zanzibar | 9-30 | 39.1408 | -6.135 | 64 | 308 | fringing reef |
| Stone Town | Zanzibar | 12 | 39.1474 | -6.2137 | 77 | 519 | fringing reef |
| Stone Town | Zanzibar | 20 | 39.1474 | -6.2137 | 66 | 361 | fringing reef |
| Menai Bay | Zanzibar | 1 | 39.3719 | -6.3236 | 36 | 241 | mangrove |
| Kizimkazi Beach | Zanzibar | 10 | 39.46 | -6.4381 | 24 | 59 | sand flat |
| Mafia outside | Mafia | 21 | 39.828 | -7.9179 | 44 | 94 | fore reef |
| Mafia outside | Mafia | 20 | 39.8224 | -7.9221 | 82 | 347 | fore reef |
| Chole Bay 1 | Mafia | 18-21 | 39.8173 | -7.9414 | 27 | 74 | backreef |
| Chole Bay 2 | Mafia | 15-18 | 39.7871 | -7.9483 | 77 | 241 | fringing reef |
| Chole Bay 2 | Mafia | 20 | 39.786 | -7.9491 | 72 | 281 | fringing reef |
| Mafia Lodge | Mafia | 0-3 | 39.7479 | -7.9734 | 62 | 397 | fringing reef |


3.2 Quantitative analysis
We used Hill numbers (i.e., the effective number of equally abundant species)
parameterized by a diversity order $q$ to estimate ostracod diversity in each sample and
island (Hill, 1973). Hill numbers have several major advantages over other diversity
indices and are increasingly adopted by ecologists (Chao et al., 2020). For example, the
Hill numbers will double when combining two identically distributed but distinct
communities, so they obey the "doubling property" and behave like species richness
(Chao et al., 2014b). In other words, the unit of Hill numbers is also "species" and thus
is more ecologically meaningful than other traditional diversity indices. Also, the order
$q$ of the Hill numbers controls the sensitivity of the diversity metric to species relative
abundance. When the order $q=0$, Hill number ($^{0}D$) reduces to species richness; when
the order $q=1$, Hill number ($^{1}D$) measures the diversity of the abundant species; when
the order $q=2$, Hill number ($^{2}D$) measures the diversity of dominant species (Chao et
al., 2014b). Therefore, besides species richness, the Hill numbers also estimate the
effective (or hypothetical) numbers of abundant and dominant species. Coincidentally,
the Hill numbers $^{1}D$ and $^{2}D$ are equivalent to the exponential of Shannon entropy and
Simpson index (hereafter referred to as Shannon and Simpson diversity), respectively



(Chao et al., 2014b), making them conceptually easy to understand by ecologists. To
make a fair comparison among multiple assemblages, we standardized the Hill numbers
with rarefaction or extrapolation to the largest sample completeness possible across
samples (82.5%) and across islands (98.6%) (Chao et al., 2020). The standard error and
95% confidence intervals of the Hill numbers were estimated by bootstrap resampling,
which was repeated 1000 times. Species evenness, $^qE_3(p) = (^qD − 1)/(S − 1)$, where $^qD$
denotes Hill numbers of order q, and $S$ denotes species richness, was quantified using
the continuous profiles of Hill numbers as functions of order $q$ (Chao and Ricotta, 2019).
A gradual profile suggests a more even community in which the species richness and
number of abundant and dominant species are similar. In contrast, a steep profile
indicates an uneven community comprised of one or a few dominant species (Mamo et
al., 2023).

To distinguish biofacies associated with different benthic habitats, we conducted
hierarchical cluster analysis based on Ward's minimum variance and three Hill number-
based dissimilarity indices, including Sørensen ($q$=0), Horn ($q$=1), and Morisita-Horn
($q$=2), to estimate the effective proportion of un-shared species in the ostracod
assemblages (Chao et al., 2014a). Similarly, the order $q$ controls the sensitivity of the
Hill number-based dissimilarities to species relative abundance. While the classic
Sørensen dissimilarity is presence-absence based, the latter two indices are designed to
quantify the compositional dissimilarities of abundant and dominant species,
respectively. The Ward's algorithm is preferred for delineating biofacies because it
minimizes the error sum of squares within clusters and generates more balanced clusters.
The number of clusters was determined by considering both the structure of the
dendrograms and the average silhouette width, with a higher value indicating greater
cohesion and separation of clusters. We also performed a non-Metric Multidimensional
Scaling (nMDS) to visualize and summarize faunal similarities among ostracod
assemblages in two-dimensional space. Stress values were calculated to quantitatively
weigh the 'goodness of fit' between the original input data matrix and the ultrametric
matrix of the resultant nMDS scatter plots (Hong et al., 2022; Kruskal, 1964). We used
a compositional heat map to illustrate the relationships between samples by Horn
dissimilarities and between species by Hellenger distances.



All analyses were implemented in RStudio. We used the package 'iNEXT' to estimate
diversity (Chao et al., 2014a; Hsieh et al., 2016) and 'vegan' for our multivariate
analyses (Oksanen et al., 2020). Figures and maps were constructed using 'ggplot2'
(Wickham, 2020).

**4 Results**
4. 1 Diversity
A total of 6262 ostracods were recovered from 26 samples at 16 locations around the
Zanzibar Archipelago. They represent remarkably diverse ostracod assemblages
comprised of 235 species under 77 genera. Considering the alpha diversity of individual
sample as measured by Hill number of different order $q$, the spatial diversity patterns
were relatively consistent for rare (i.e., species richness, $^{0}$D) and abundant ($^{1}$D) species.
The highest values were recorded for fringing reefs at Chole Bay 2 and Ras Nungwi,
followed by fringing reefs at Mafia outside and Ras Nungwi peak (Figs. 2A, 3, S1-S2).
Moderately high levels of diversity were observed at fore reef sites in Pemba Island and
fringing reefs at Bawe, Stone Town, and Mafia Lodge. In terms of the diversity of
dominant ($^{2}$D) species, there was a more homogenous distribution with similarly high
values found at various fringing and fore reefs, including Chole Bay 2, Mafia outside,
Haramu Passage, Bawe Island, Ras Nungwi and Ras Nungwi peak. All remaining
localities (Chole Bay 1, Mnemba Atoll, Ocean Paradise and Kizimkazi Beach)
characterized by sand flat and back reef habitats had consistently low diversity across
all order $q$, especially Menai Bay that was lined with mangrove stands. Evenness was
highest at Haramu Passage and lowest at Menai Bay for both orders $q$=1 and $q$=2 (Figs.
2B, S3). With respect to the gamma diversity of each island, Mafia and Zanzibar were
almost equally diverse across all order $q$, while Pemba had significantly lower diversity
for abundant and dominant species (Figs. 2C, S4).




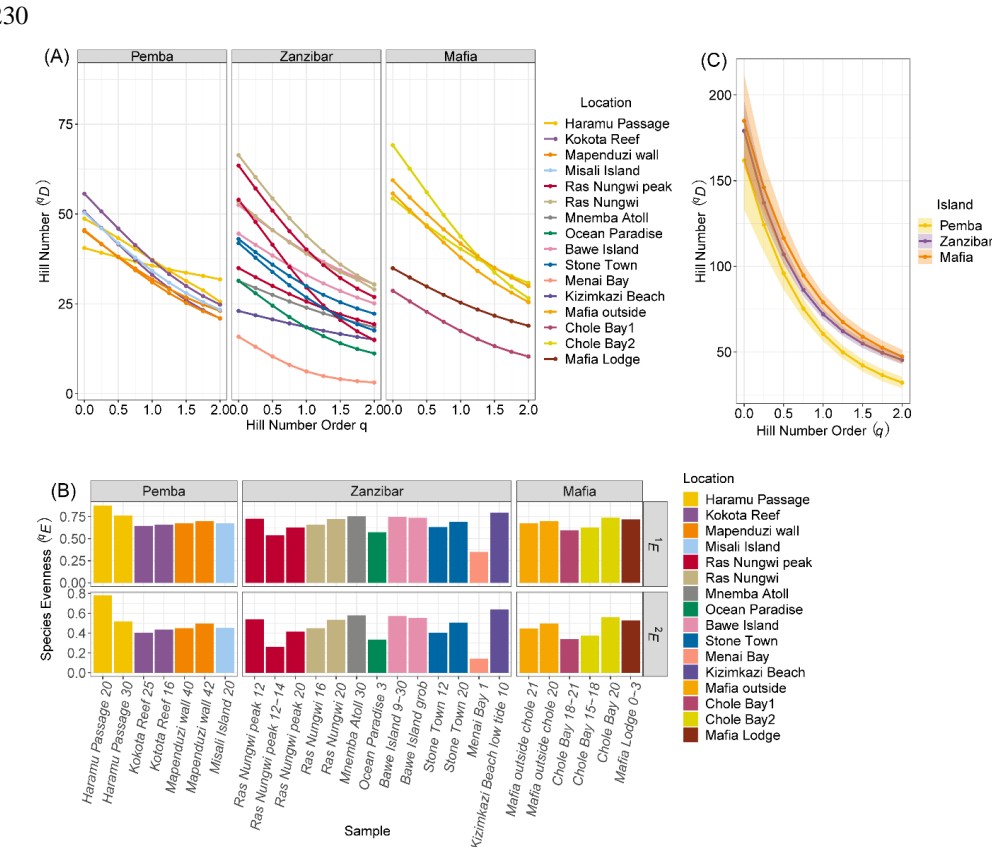


Fig. 2. Diversity results of Zanzibar Archipelago ostracods. (A) Alpha diversity of each
sample shown by Hill number profile based on 82.5% sample coverage. The overall
elevation of the profile indicates the diversity based on hill number across different
order $q$. The levelness of the line indicates species evenness of the assemblage, because
a complete leveled diversity profile would suggest that the numbers of total, common
and dominant species are all the same. (B) Evenness of each sample as the normalized
slope of Hill number profile for order $q=1$ and $q=2$ based on 82.5% sample coverage.
(C) Gamma diversity of each island shown by Hill number profile based on 98.6%
sample coverage. The shade area shows 95% confidence interval of the profile.



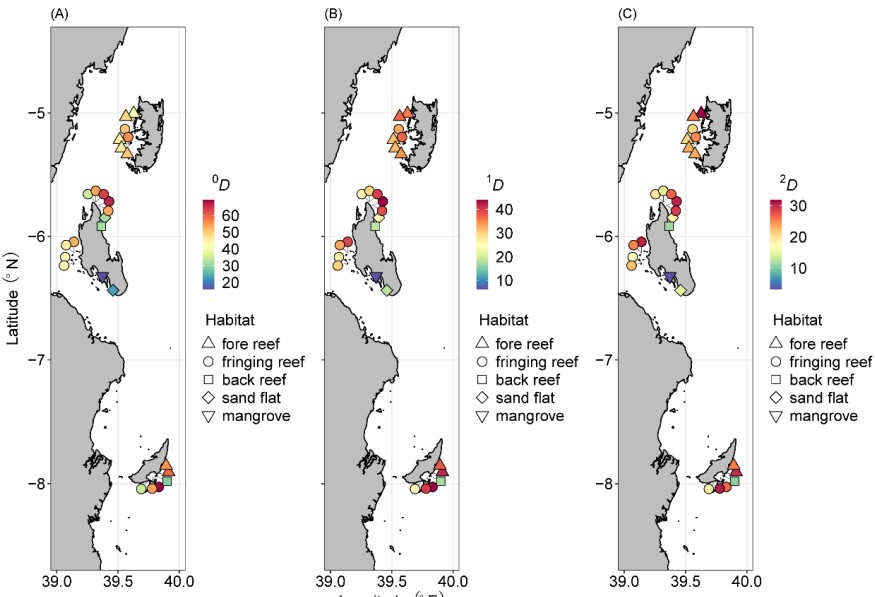


243 Fig. 3. Diversity maps of Zanzibar archipelago ostracod. Distributions of Hill numbers

244 $^0D$ (A: $q = 0$, species richness), $^1D$ (B: $q = 1$, exponential Shannon), and $^2D$ (C: $q = 2$,

245 inversed Simpson). We used 82.5% sample coverage to standardize the Hill number

246 estimates. Diversity and habitat are represented by color and shape as in the legends,

247 respectively.

248

249 4.2 Multivariate analysis

250 First, cluster analyses based on Sørensen, Horn, and Morisita-Horn dissimilarities

251 delineated biofacies considering faunal composition in terms of species occurrence,

252 relative abundance of abundant species, and relative abundance of dominant species,

253 respectively. The greatest average silhouette width suggested the division of samples

254 into ten clusters for all three dissimilarity measures; however, it is beyond interpretable

255 to have too many clusters, given the size of our dataset. We, therefore, referred to the

256 structure of the dendrograms based on three dissimilarity measures to determine the

257 optimum number of clusters to be four (Fig. S5). The NMDS results showed a clear

258 separation of four biofacies based on Horn and Morisita-Horn dissimilarities, but not

259 Sørensen dissimilarity, which was calculated with a relatively high stress value (0.26)

260 (Fig. 4). Ostracod faunas in Pemba Island constituted a distinct group across all levels

261 of faunal composition from presence/absence to relative abundance (Biofacies 1; Fig.





5). Ras Nungwi, Ras Nungwi peak, and nearby Menemba Atoll were congregated with
different sites around Zanzibar and Mafia in Biofacies 2, including Mafia outside and
Chole Bay 2 in Sørensen, Mafia outside, Chole Bay 1 and Chole Bay 2 in Horn, Ocean
Paradise, Kizimkazi Beach and Mafia Lodge in Morisita-Horn analysis (Fig. 5).
Samples assigned to Biofacies 3 and 4 strongly varied depending on the dissimilarity
matrix used, indicating these biofacies have different ecological meaning among three
cluster analyses. Specifically, they scattered around the entire Zanzibar Island based on
Sørensen dissimilarity. Biofacies 4 was distributed along the western coast of Zanzibar,
including Stone Town and Bawe, and Biofacies 3 covered the remaining Zanzibar
locations (Menai Bay, Ocean Paradise and Kizimkazi Beach) in addition to Mafia
Lodge based on Horn dissimilarity. On the other hand, when Morisita-Horn
dissimilarity was applied, Menai Bay was different from all other sites as a distinctive
Biofacies 3 while most Mafia sites (Mafia outside, Chole Bay 1, and Chole Bay 2)
aggregated in Biofacies 4. Considering the performance of multivariate analyses to
reflect and interpret biological patterns, we think that cluster and NMDS results based
on Horn dissimilarity most reasonably captured the underlying ecological significance
of reefal versus non-reefal facies as determined by benthic community, depth, and
possibly anthropogenic disturbances (see the Discussion section). We therefore focus
on the four biofacies as divided by Horn-based analysis to scrutinize their diversity and
compositional structure in relation to a set of environmental variables.

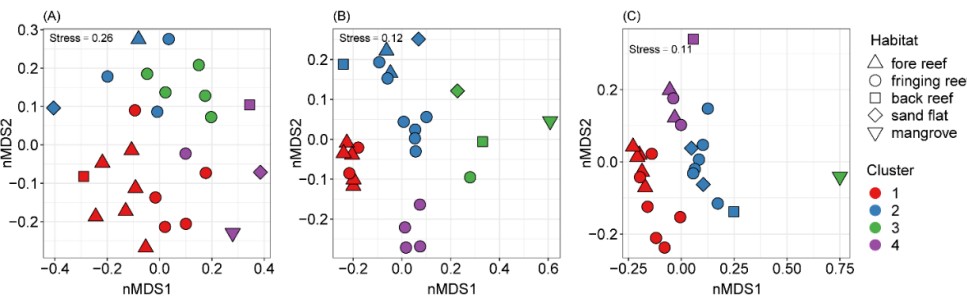


Fig. 4. nMDS ordinations showing biofacies based on (A) Søensen, (B) Horn, and (C)
Morisita-Horn dissimilarities and Ward's minimum variance cluster analysis. Cluster
and habitat are represented by color and shape as in the legends, respectively. Note that
the color schemes are independent among panels; thus, the biofacies based on different
dissimilarities are not necessarily related.

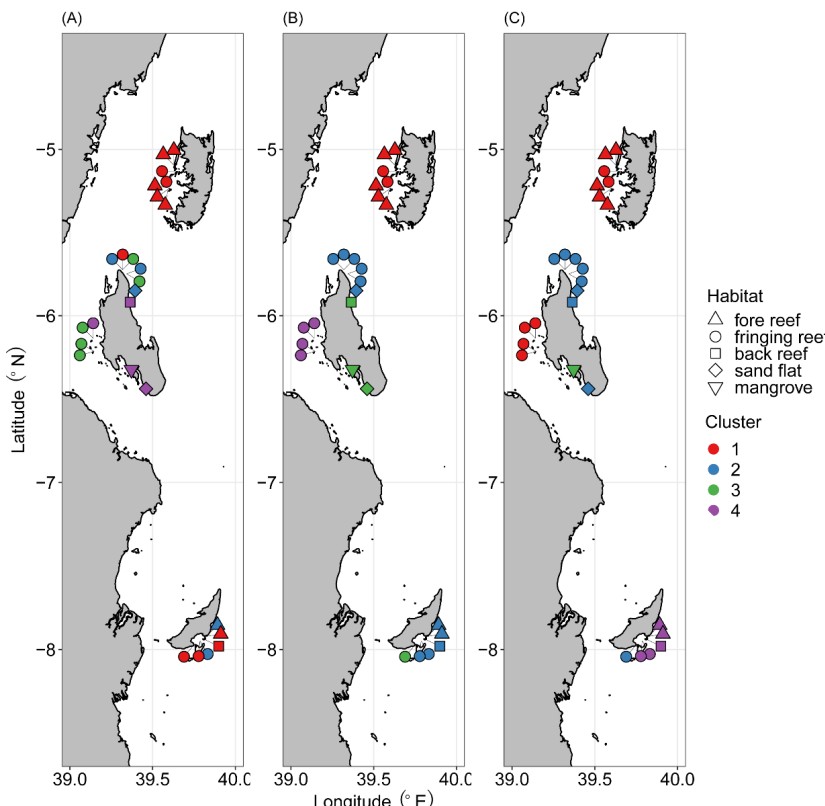

Fig. 5. Distribution of ostracod Biofacies 1-4 based on (A) Søensen, (B) Horn, and (C) Morisita-Horn dissimilarities and Ward's minimum variance cluster analysis. Note that the color schemes are independent among panels; thus, the biofacies based on different dissimilarities are unrelated. Cluster and habitat are represented by color and shape as in the legends, respectively.

Each biofacies based on Horn dissimilarity index was demonstrated with the top 10 species of highest mean relative abundance as shown in Table 2 and Figures 6-8. Noticeably, the Pemba fauna in Biofacies 1 was dominated by genus *Neonesidea* (*N.* cf. *crepidula* and *N. schulzi*) and *Bosasella* (*B. profunda* and *B. elongate*), together with *Paracytheridea tschoppi* (Fig. 9; Table 2). Biofacies 2 included the most diverse sites in Zanzibar and Mafia, which all shared similar faunal structures with a high abundance of *Loxocorniculum* sp. 2, *Xestoleberis rotunda, Paracytheridea albatros* and *Loxoconcha* sp. 3. Biofacies 3 composed of low-diversity sites in Zanzibar and Mafia was distinguished by highly abundant *Perissocytheridea* sp.1, *Xestoleberis hanaii*, as




well as three *Loxoconcha* species (*L.* sp. 3, *L. ghardaqensis* and *L. lilljeborgii*). Finally,
the faunal structure of Biofacies 4 in western Zanzibar showed some similarities to that
of Biofacies 1 in Pemba with many common species, however, they clearly differed by
the dominance of *Xestoleberis hanaii* and *Patrizia nucleuspersici* in Biofacies 4.

Table 2. List of top 10 species of highest % mean relative abundance for Biofacies 1-4
based on Horn dissimilarity.

| Species | Biofacies1 | Biofacies2 | Biofacies3 | Biofacies4 |
|---|---|---|---|---|
| *Neonesidea* cf. *crepidula* | 0.085857 | NA | NA | NA |
| *Bosasella profunda* | 0.079436 | NA | NA | 0.040846 |
| *Neonesidea schulzi* | 0.075285 | 0.032551 | 0.024322 | 0.041291 |
| *Paracytheridea tschoppi* | 0.035779 | NA | NA | 0.028826 |
| *Loxocorniculum* sp. 2 | 0.030562 | 0.063399 | NA | NA |
| *Xestoleberis hanaii* | 0.028593 | 0.039954 | 0.084378 | 0.071834 |
| *Patrizia nucleuspersici* | 0.02842 | NA | NA | 0.057965 |
| *Paranesidea* cf. *spongicola* | 0.026203 | NA | NA | 0.029754 |
| *Xestoleberis* sp. 1 | 0.023801 | NA | NA | NA |
| *Bosasella elongata* | 0.023369 | NA | 0.017579 | NA |
| *Xestoleberis rotunda* | NA | 0.061861 | NA | NA |
| *Paracytheridea albatros* | NA | 0.045056 | 0.037464 | NA |
| *Loxoconcha* sp. 3 | NA | 0.041327 | 0.110386 | NA |
| *Bosasella* sp. 1 | NA | 0.040122 | NA | NA |
| *Macrocyprina maddocksae* | NA | 0.039264 | NA | NA |
| *Caudites exmouthensis* | NA | 0.027832 | NA | NA |
| *Paranesidea* sp. 1 | NA | 0.025497 | NA | NA |
| *Perissocytheridea* sp.1 | NA | NA | 0.157932 | NA |
| *Loxoconcha ghardaqensis* | NA | NA | 0.073153 | NA |
| *Hiltermannicythere rubrimaris* | NA | NA | 0.04805 | NA |
| *Loxoconcha lilljeborgii* | NA | NA | 0.033061 | NA |
| *Neohornibrookella lactea* | NA | NA | 0.018616 | NA |
| *Neonesidea* sp. 3 | NA | NA | NA | 0.048331 |
| *Neonesidea paiki* | NA | NA | NA | 0.042016 |
| *Loxoconcha* cf. *gisellae* | NA | NA | NA | 0.035319 |
| *Perissocytheridea?* sp. 2 | NA | NA | NA | 0.029391 |




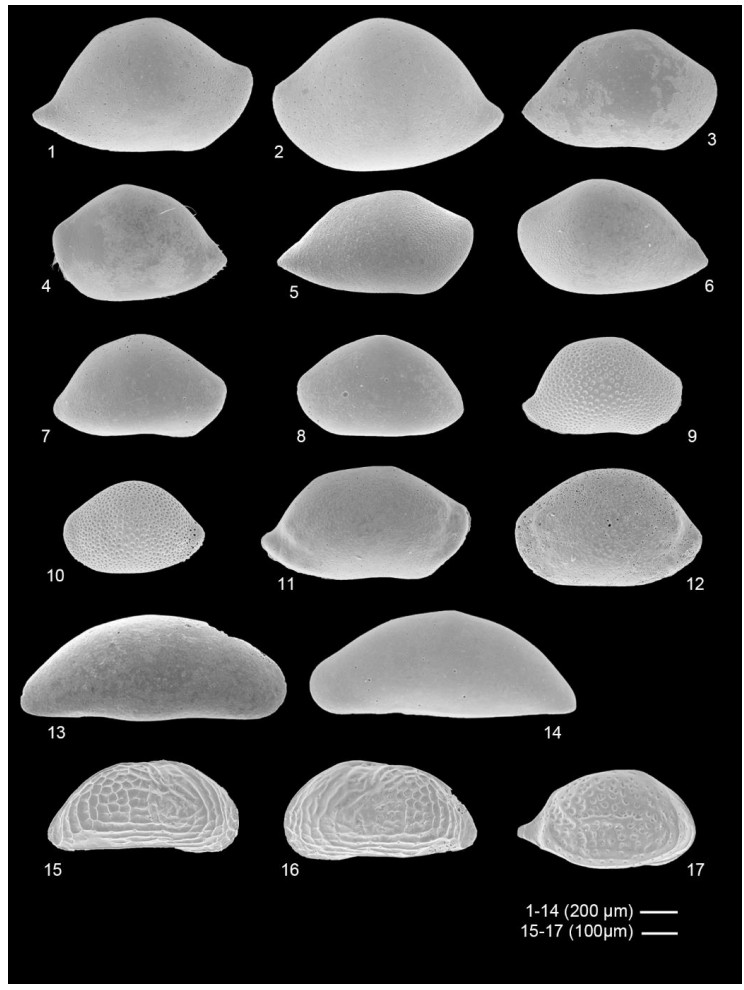


Fig. 6. Scanning electron microscopy images of the top 10 ostracod species of highest % mean relative abundance for Biofacies 1-4 based on Horn dissimilarity. 1, *Neonesidea* cf. *crepidula*, RV; 2, *Neonesidea* cf. *crepidula*, LV; 3, *Neonesidea paiki*, RV; 4, *Neonesidea paiki*, LV; 5, *Neonesidea schulzi*, RV; 6, *Neonesidea schulzi*, LV; 7, *Neonesidea* sp. 3, RV; 8, *Neonesidea* sp. 3, LV; 9, *Paranesidea* cf. *spongicola*, RV; 10, *Paranesidea* cf. *spongicola*, LV; 11, *Paranesidea* sp. 1, RV; 12, *Paranesidea* sp. 1, LV; 13, *Macrocyprina maddocksae*, RV; 14, *Macrocyprina maddocksae*, LV; 15, *Perissocytheridea* sp.1, RV; 16, *Perissocytheridea* sp.1, LV; 17, *Perissocytheridea*? sp. 2, RV. All adults and lateral views.

323



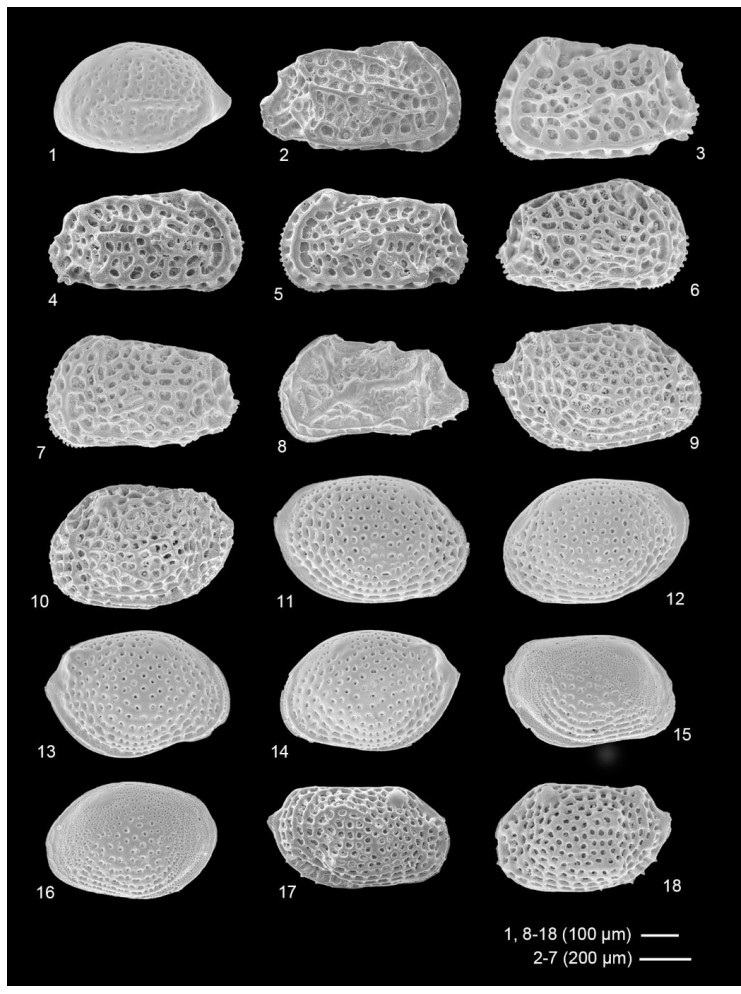

324

Fig. 7. Scanning electron microscopy images of the top 10 ostracod species of highest % mean relative abundance for Biofacies 1-4 based on Horn dissimilarity. 1, *Perissocytheridea*? sp. 2, LV; 2, *Bosasella elongate*, RV; 3, *Bosasella elongate*, LV; 4, *Bosasella profunda*, RV; 5, *Bosasella profunda*, LV; 6, *Bosasella* sp. 1, RV; 7, *Bosasella* sp. 1, LV; 8, *Caudites exmouthensis*, LV; 9, *Loxoconcha ghardaqensis*, RV; 10, *Loxoconcha ghardaqensis*, LV; 11, *Loxoconcha* cf. *gisellae*, RV; 12, *Loxoconcha* cf. *gisellae*, LV; 13, *Loxoconcha lilljeborgii*, RV; 14, *Loxoconcha lilljeborgii*, LV; 15, *Loxoconcha* sp. 3, RV; 16, *Loxoconcha* sp. 3, LV; 17, *Loxocorniculum* sp. 2, RV; 18, *Loxocorniculum* sp. 2, LV. All adults and lateral views.

334



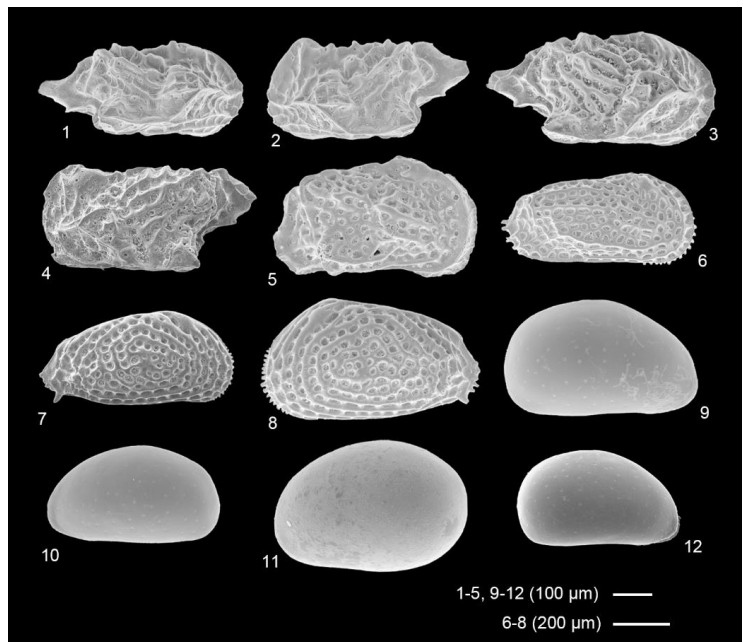

Fig. 8. Scanning electron microscopy images of the top 10 ostracod species of highest % mean relative abundance for Biofacies 1-4 based on Horn dissimilarity. 1, *Paracytheridea albatross*, RV; 2, *Paracytheridea albatross*, LV; 3, *Paracytheridea tschoppi*, RV; 4, *Paracytheridea tschoppi*, LV; 5, *Neohornibrookella lactea*, RV; 6, *Hiltermannicythere rubrimaris*, RV; 7, *Patrizia nucleuspersici*, RV; 8, *Patrizia nucleuspersici*, LV; 9, *Xestoleberis hanaii*, RV; 10, *Xestoleberis hanaii*, LV; 11, *Xestoleberis rotunda*, LV; 12, *Xestoleberis* sp. 1, RV. All adults and lateral views.



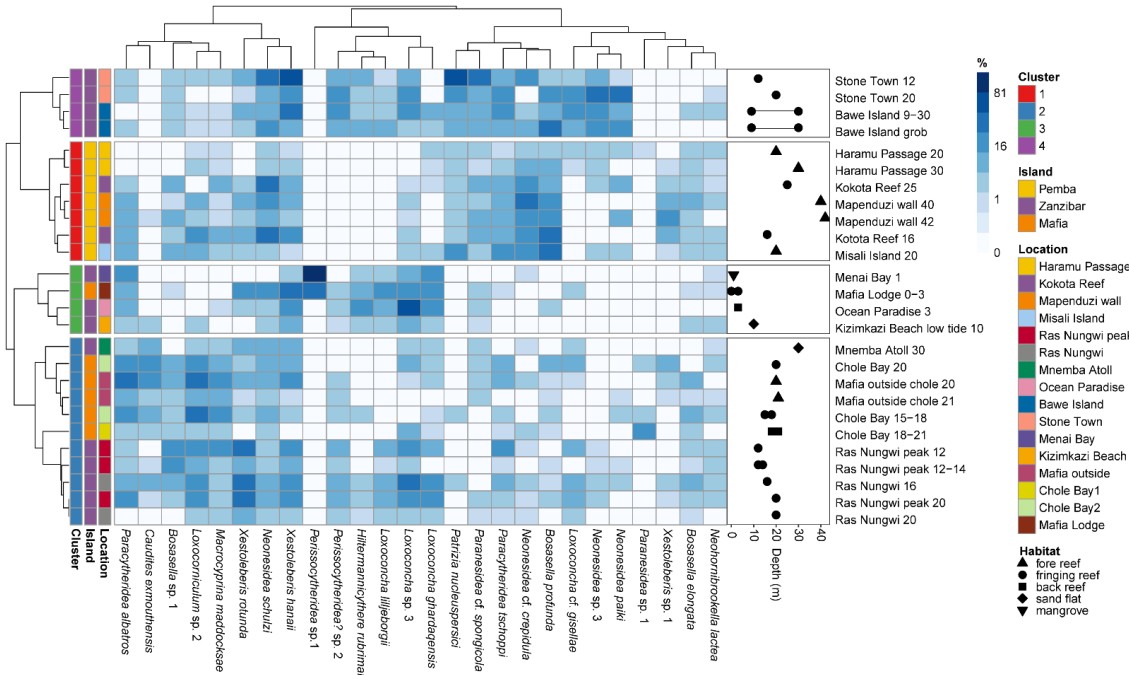


Fig. 9. Dendrograms based on Horn dissimilarity between samples and Hellenger
distances between top 10 species of highest mean relative abundance in each cluster.
The blue heatmap indicates the relative (%) abundance of each species in each sample.
The side panel shows water depth and habitat type of each sample (note that several
samples are shown by their corresponding depth ranges).

**5 Discussion**
Through Hill number profile and multivariate analyses, we quantified a highly diverse
ostracod fauna in the Zanzibar Archipelago composed of four distinct biofacies. The
delineation of biofacies varied considerably depending on the dissimilarity matrix used,
indicating inconsistent faunal structures across different levels of species information
from occurrence to relative abundance (Fig. 5). In terms of the presence/absence of
species (Sørensen dissimilarity), all Pemba sites united in Biofacies 1 but the
assignment of Zanzibar and Mafia sites into Biofacies 1-4 seemingly conformed to a
noisy pattern. Accordingly, four biofacies intersected with each other in nMDS space
with relatively high stress value (Fig. 4A). A possible explanation is that the occurrence
of individual species may be homogenous among sites in similar environmental

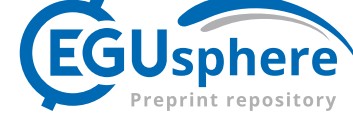

conditions within a finite geographic region. Many species are likely to be ubiquitous
across the entire neritic zone despite showing certain ecological preferences, and the
redeposition processes may further facilitate the mixing of death assemblages to blur
the spatial signal at a local scale (Frenzel and Boomer, 2005). Consequently, species
presence in all available habitats may translate to considerable faunal similarities
among biofacies as measured by Sørensen index. When considering the composition of
abundant species (Horn dissimilarity) (Figs. 4B and 5B), the identification of four
biofacies instead reflected significant changes in ostracod assemblages along two
important environmental gradients, which are benthic community type and water depth.
Specifically, Biofacies 1 and 2 characterize typical fore reefs in deep subtidal (sampling
depth 16-42 m) and fringing reefs in shallow subtidal (12-30 m), respectively (Fig. 9).
Biofacies 3 indicates intertidal habitats with plant cover (0-10 m), and finally Biofacies
4 features degraded fringing reefs in shallow subtidal (9-30 m) (see discussion below).

We summarized the ecological preferences of dominant genera in each biofacies based
on Horn dissimilarity (Table 3 and Fig. 9) and investigated how key environmental
factors (benthic community type, water depth, and anthropogenic disturbance) may
control the distribution and diversity of reefal ostracod assemblages. First of all,
*Neonesidea* and *Paranesidea* (family Bairdiidae) are typical reefal genera that reach
their maximum diversity and incidence on reefs and reef-associated habitats in tropical
shallow-marine environments (Whatley and Watson, 1988; Maddocks, 2013; Titterton
and Whatley, 1988). Their dominance in Biofacies 1 is consistent with our background
understanding that the Pemba reefs were pristine and healthy (Ussi et al., 2019;
Grimsditch et al., 2009). However, it should be noted that individual species of these
genera likely have different environmental tolerance. For example, *N.* cf. *crepidula*
were restricted to Biofacies 1 while *N. schulzi* were widespread among four biofacies
inhabiting both reef and algae habitats (Fig. 9) (Mostafawi et al., 2005). *Bosasella* as
another prominent component of Biofacies 1 is also known to occur on coral reefs in
the western Indian Ocean (Munef et al., 2012; Jellinek, 1993). *Paracytheridea* and
*Caudites* on the other hand are loosely categorized as reefal genera, as their dominance
on coral reefs was reported but not studied in detail (Whatley and Watson, 1988; Keyser
and Mohammed, 2021). In this study, they were common on fore- and fringing-reefs in
Biofacies 1 and 2 (Fig. 9). *Loxoconcha* and *Loxocorniculum* (family Loxoconchidae)
as two phylogenetically related and ecologically similar genera exhibited ubiquitous



distribution around Zanzibar Archipelago with highest relative abundance in Biofacies
3 followed by Biofacies 2. As generalists, they thrive on a wide variety of benthic
habitats across the neritic zone and show affinities to plant substrates (algae and
seagrass beds) in particular (Munef et al., 2012; Keyser and Mohammed, 2021; Kamiya,
1988). The ecology of *Xestoleberis* is very similar to that of Loxoconchidae, living both
on coral reefs and algal flats (Keyser and Mohammed, 2021; Munef et al., 2012;
Whatley and Watson, 1988; Kamiya, 1988). This genus was almost equally weighted
in all biofacies, although individual species clearly preferred different environments, as
*X. hanaii* prevailed in Biofacies 3 and 4 while *X. rotunda* only in Biofacies 2 (Fig. 9).
*Patrizia* is documented as a reefal genus in lower littoral zone along the eastern coast
of tropical Africa (Jellinek, 1993). It dominated the relatively deep fringing-reef faunas
of Biofacies 4,  which were subject to sewage-derived nutrient and trace metal pollution
from the Zanzibar Town (Narayan et al., 2022; Bravo et al., 2021). Different from all
the above-discussed genera, *Hiltermannicythere* and *Perissocytheridea* are restricted to
shallow intertidal environments as phytal and sediment-dwelling taxa, respectively
(Jellinek, 1993), which explains their abundance in our Biofacies 3. *Perissocytheridea*
is especially considered a bioindicator of brackish water facies (Nogueira and Ramos,
2016; Keyser, 1977). Furthermore, we revealed a more generalized pattern of the
compositional differences among biofacies with the top 5 families of highest mean
relative abundance in each biofacies (Fig. 10).

Table 3. Autoecology summary of important ostracod genera.

| Genus | Predominant habitats | References |
| --- | --- | --- |
| *Neonesidea* | Coral reef | Whatley and Watson (1988); Maddocks (2013); Titterton and Whatley (1988); Maddocks (1969) |
| *Paranesidea* | Coral reef | Titterton and Whatley (1988); Whatley and Watson (1988); Maddocks (1969) |
| *Bosasella* | Coral reef | Munef et al. (2012) |
| *Loxoconcha* | Algal mat and reef | Keyser and Mohammed (2021); Whatley and Watson (1988); Munef et al. (2012); Kamiya (1988) |
| *Loxocorniculum* | Algal mat and reef | Munef et al. (2012); Kamiya (1988) |
| *Xestoleberis* | Algal mat and reef | Keyser and Mohammed (2021); Whatley and Watson (1988); Munef et al. (2012); Kamiya (1988) |
| *Patrizia* | Coral reef | Jellinek (1993) |



| *Hiltermannicythere* | Intertidal algal mat | Jellinek (1993); Keyser and Mohammed (2021) |
| *Paracytheridea* | Coral reef | Whatley and Watson (1988) |
| *Caudites* | Coral reef | Whatley and Watson (1988); Keyser and Mohammed (2021) |
| *Perissocytheridea* | Intertidal sand flat, brackish water | Nogueira and Ramos (2016); Keyser (1977) |



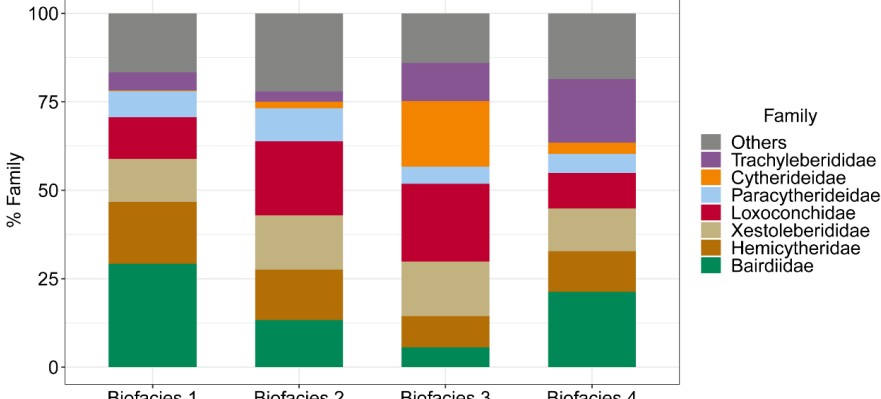


Fig. 10. Family composition of Biofacies 1-4 based on Horn dissimilarity. The top 5
families of the highest % relative abundance in each biofacies are shown.

Thus, the distribution of shallow-marine ostracods in the Zanzibar Archipelago is
characterized by three reefal facies and one intertidal facies. Yet slight differences in
bathymetry, benthic community type, and anthropogenic impacts likely contributed to
subtle faunal changes among the reefal Biofacies 1, 2, and 4. The fore reefs in Pemba
(Biofacies 1) were deepest with high incidence and diversity of live hard corals
(Gavrilets and Losos, 2009; Ussi et al., 2019), which accounted for the definite
dominance of ostracod reefal taxa (Bairdiidae and *Bosasella*) over algal taxa
(Loxoconchidae and Xestoleberididae) (Figs. 9-10). Moderately high levels of diversity
in terms of rare, abundant, and dominant species were observed for these ostracod
assemblages (Figs. 2-3). The Pemba reefs are thereby considered the most mature and
authentic reef ecosystem, serving as a natural reference for comparing with other sites.
The fringing-reef fauna of western Zanzibar (Stone Town and Bawe, Biofacies 4)
exhibited certain similarities with the Pemba fauna as indicated by the prevalence of



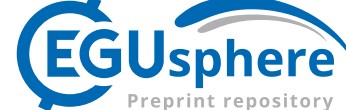

*Bosasella profunda*, *Paracytheridea tschoppi*, and *Paranesidea* cf. *spongicola* in both
facies (Fig. 9). Indeed, they were grouped together based on the composition of
dominant species (Morisita-Horn analysis) (Fig. 5C). Faunal similarities between
Pemba and Stone Town make sense as they are in comparable baseline conditions of
water depths and hydrology along the protected western coast of Zanzibar Archipelago,
in contrast to Ras Nungwi and Chole Bay that are exposed to oceanic disturbances from
the east (Fig. 1). However, Biofacies 4 was differentiated from Biofacies 1 by the
dominance of *Patrizia* in conjunction with the absence of *Neonesidea* cf. *crepidula*. It
also had the highest relative abundance of Trachyleberididae genera among all facies
(Fig. 10), for example, *Adencythere*, *Strobilocythere*, *Bradyon*, and *Actinocythereis*, but
their ecologies are not well understood. Stressful environmental conditions in terms of
overexploitation, tourism and coastal pollution offer the most possible explanation for
such a unique faunal composition and comparatively low diversity of Biofacies 4 (Figs.
2-3) (Bravo et al., 2021; Larsen et al., 2023). Consistently, foraminifera and coral
surveys indicated early stages of reef degradation there (Narayan et al., 2022; Bravo et
al., 2021; Thissen and Langer, 2017) It is possible that ongoing anthropogenic
disturbances near the Stone Town will eventually exceed the critical threshold levels to
cause more pronounced changes in ostracod faunal structures in terms of dominant
species through a shift in benthic habitat (Narayan et al., 2022; Hong et al., 2022). Other
than Biofacies 1 and 4, Biofacies 2 represented a different type of reefal habitat of Ras
Nungwi, Chole Bay, Mafia outside, and Mnemba Atoll (Fig. 5B). Algal taxa
(Loxoconchidae and Xestoleberididae) and reefal taxa (Bairdiidae, *Bosasella*,
*Paracytheridea*, and *Caudites*) reached equally high levels of relative abundance there
(Figs. 9-10). Most sites in Biofacies 2 were relatively shallow (12-21 m) except for
Mnemba Atoll (30 m), and they covered the transitional zone from intertidal sandy
bottom to subtidal true reefs. Microhabitats on the reef platforms of Biofacies 2 are
believed to be diverse and heterogenous with interlaced live and dead corals, algae and
seagrass, calcareous sands, as well as bare substrate rock (Ussi et al., 2019; Larsen et
al., 2023), which facilitated the coexistence of reefal and algal ostracods and
consequently the highest diversity of local assemblages (Figs. 2-3). The remaining
Biofacies 3 corresponded to the shallowest intertidal habitats with various benthic
communities, including back reef, fringing reef, sand flat, and mangrove (Fig. 5B).
Typical reefal taxa (Bairdiidae and *Bosasella*) dropped to their lowest relative
abundance in this facies, replaced by large numbers of Loxoconchidae,



*Perissocytheridea,* and *Hiltermannicythere* that well adapted to shallow euryhaline
conditions (Figs.9-10). Not surprisingly, the diversity of Biofacies 3 was much lower
than that of open-ocean reefal facies, as drastic changes in temperature, salinity,
dissolved oxygen, and wave energy in the intertidal zone may be too challenging for
many marine taxa (Figs. 2-3) (Morley and Hayward, 2007; Frenzel and Boomer, 2005).
The mangrove habitat at Menai Bay was unique concerning the absolute dominance of
*Perissocytheridea* in line with its lowest diversity and evenness (Figs. 2-3). It indeed
constituted an independent biofacies based on Morisita-Horn analysis (Fig. 5C).
The division scheme of four biofacies based on Horn dissimilarity explicitly revealed
spatial patterns of ostracod distribution in aspect of diversity and composition, as
discussed above. Our results are generally concordant with a previous study on benthic
foraminifera, which separated six clusters of Pemba, Stone Town, Mafia Bay, Ras
Nungwi, Mnemba Atoll, and Menai Bay, respectively (Fig. 11B) (Thissen and Langer,
2017). Each of these foraminifera clusters corresponded to major habitat types, as
argued by the authors (Thissen and Langer, 2017), and we accordingly pointed out the
consistent role of habitat factors in shaping the biogeography of both ostracod and
foraminifera biotas. However, the diversity patterns of these two groups were
apparently different among reefal habitats (Figs. 3, 11A). High, moderate, and low
levels of diversity were recorded on fore reefs (Pemba), fringing reefs (Mafia and
Zanzibar), and intertidal (Zanzibar) for foraminifera, in contrast to fringing reefs (Mafia
and Zanzibar), fore reefs (Pemba), and intertidal (Zanzibar) for ostracods, respectively.
Such discrepancies may imply a tight association of foraminifera with reef ecosystem
and their ultra-sensitivity to reef health, since their diversity generally decreased from
pristine, mature reefs to degraded, marginal reefs. Ostracods, on the other hand, may
be less confined or specific to reef habitats. The occupation of coral and algae substrate
by distinct faunal groups allows them to thrive in the transitional zone between marginal
and true reefs.



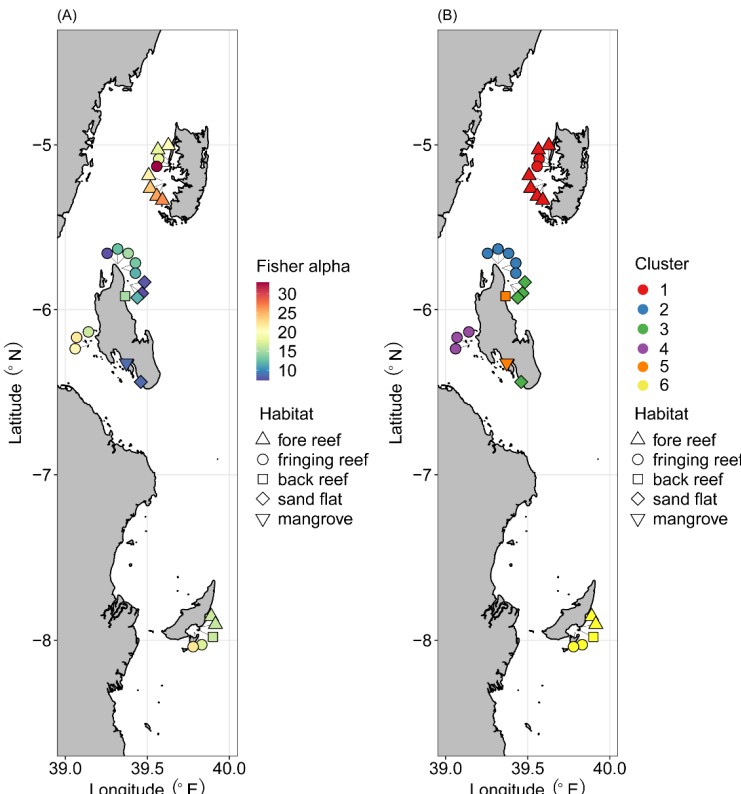


Fig. 11. Distributions of benthic foraminifera (A) diversity measured as Fisher alpha
index; (B) cluster groups based on Q-mode cluster analysis. Modified from Thissen and
Langer (2017). Diversity/cluster and habitat are represented by color and shape as in
the legends, respectively.

504

Most importantly, this study established a clear benthic community axis along which
the composition and diversity of ostracod assemblage vary, i.e., from coral reefs to
algae turfs. We identified typical reefal association (Bairdiidae-*Bosasella*) versus algal
association (Loxoconchidae-Xestoleberididae) (Fig. 10), and their relative dominance
may be used as a direct indication of benthic community type. As there is a growing
interest to monitor the degradation of reef ecosystems from the coral-dominated phase
to the algae-dominated phase (Roth et al., 2018; Knowlton and Jackson, 2008;
Knowlton, 2012), our finding is of potential conservation value. Ostracod species
diversity was higher on shallow fringing reefs than on deep fore reefs, as the former
ecosystem harbored evenly weighted reefal and algal taxa within a dynamic mosaic of



microhabitats. Our results thus strongly indicate the importance of coral reefs in
harboring conspicuously high levels of meiobenthic biodiversity, likely through finer
niche partitioning (Kohn et al., 1997; Fox and Bellwood, 2013). Along with the benthic
community factor, we quantified prominent changes in faunal structure and diversity
along a depth gradient, as the intertidal euryhaline assemblages transited to subtidal
fully marine assemblages. It is widely recognized that shallow-marine biotas are
especially susceptible to depth associated changes, such as temperature, salinity, wave
action, and light penetration (Carvalho et al., 2012; Tian et al., 2022). This study
showed that a narrow depth zone across the intertidal and subtidal (~40 m) was further
divided and occupied by distinct biofacies. Such a finely tuned vertical gradient of
diversity and faunal composition added to an exceedingly large regional species pool
(235 species) in this tropical shallow-marine setting. Last but not least, it should be
aware that the effects of depth and benthic community type are often intertwined with
each other in determining ostracod assemblages, as the habitat-building corals and algae
essentially exhibit depth distributions. At a regional scale like the Zanzibar Archipelago,
the combined effects of water depth and benthic community characteristics should be
considered in studying the spatial patterns of benthic organisms.

**6 Conclusion**
In conclusion, this study showed that the diversity and faunal composition of reefal
ostracod assemblages vary along benthic community and bathymetric gradients, which
may also be altered by local anthropogenic disturbances. Ostracod faunas on shallow
fringing reefs were especially diverse, which may be explained by high levels of habitat
complexity and heterogeneity. The relative dominance of reefal taxa (Bairdiidae)
versus algal taxa (Loxoconchidae-Xestoleberididae) is likely determined by the
proportion of coral versus algae cover on the reef platforms, though more extensive
studies beyond this region are needed to confirm the universality of this pattern. Coral
reefs worldwide are vulnerable to ongoing climate changes and other human impacts at
local to global scales, and many reefal species are at risk of extinction. It is of great
importance that we inspect and understand the immense biodiversity of meiobenthos
on coral reefs as an indispensable part of the ecosystem.

**Data availability**



Ostracod census data will be deposited into The Paleobiology Database. DOI will be
added later.

**Author contributions**


Each named author has participated sufficiently in the work to take public responsibility
for the content. SYT and ML developed the concept. ML collected the samples. SYT
carried out the experiments and collected the data. SYT and CLW performed the data
analyses. SYT drafted the manuscript. ML, MY, and CLW reviewed and edited the
manuscript.

**Competing interests**


The authors declare that they have no competing interests.

**Acknowledgements**


We thank Stephanie Pietsch, Jens Thissen, Anna Weinmann, and Michael Kunert for
their help with fieldwork; Jingfang He for her help in the lab. The work described in
this paper was supported by Humboldt Research Fellowship (to SYT), a grant from the
German Science Foundation (DFG, LA 884/10-1) (to ML), grants from the Research
Grants Council of the Hong Kong Special Administrative Region, China (project codes:
HKU 17306023, G-HKU709/21) (to MY), and grants from the National Science and
Technology Council, Taiwan (project codes: NSTC 112-2611-M-002-011) (to CW).

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
