# Peer review of "Reefal ostracod assemblages from the Zanzibar Archipelago (Tanzania)"

_EGUsphere, 2024_

## Author Comment (AC1)

MS No. egusphere-2024-487

Reefal ostracod assemblages from the Zanzibar Archipelago (Tanzania)

Dear editor and reviewers,

Here we are re-submitting our manuscript No. egusphere-2024-487. We appreciate your positive decision and giving us an opportunity to re-submit our MS. Here we addressed all the comments from three reviewers in the revised MS, and point-by-point responses are shown below.

Comments from the Reviewer#1 Andreas Haas:

With their study on 'Reefal ostracod assemblages from the Zanzibar Archipelago (Tanzania)' the authors add important knowledge on a largely understudied component of the reef ecosystem. They provide a first baseline (although already impacted by anthropogenic influence) of this functional group along the shores of eastern Tanzania. Overall the study is very thoroughly conducted, the conclusions are valid, and the results presented comprehensively. This is valuable information and should be published.

In general the manuscript is written nicely, but at some parts it becomes lengthy. This is also where my main suggestion for improvement lies. While I think the manuscript is pretty much publishable in its current form, I would like to suggest to streamline it a bit. There are many figures and tables in the manuscript - are all of them necessary or could you put some in SI or combine them? For instance Figure one and table 1 feature in large parts redundant information, maybe combine them or put table 1 in supplements?

**Response: We appreciate your positive and constructive comments. We split the original Table 1 into two parts. We kept the ostracod assemblage information and characterization of benthic habitat as the main text Table 1, as required by other reviewers. We put the sample information including geographic location, water depth, and date of collection into the supplements as supplementary table 1. The new Table 1 is as follows:**

**Table 1. Ostracod assemblage information including raw species richness, number of counted individuals, and abundance per gram sediment, in addition to a characterization of benthic habitat in terms of sediment type and algae coverage in each location.**

| Sample | Species richness | No. individuals | Abundance (per g) | Habitat | Sediment type | Algae coverage |
|---|---|---|---|---|---|---|
| **Haramu Passage20** | 37 | 69 | 1.645 | fore reef | bioclastic sand | Low |
| **Haramu Passage30** | 35 | 60 | 4.212 | fore reef | bioclastic sand | Low |
| **Kokota Reef25** | 64 | 235 | 4.176 | fringing reef | bioclastic sand | Low |
| **Kokota Reef16** | 78 | 364 | 50.845 | fringing reef | bioclastic sand | Low |

| | | | | | | |
|---|---|---|---|---|---|---|
| Mapenduzi wall40 | 60 | 235 | 37.337 | fore reef | bioclastic sand | Low |
| Mapenduzi wall42 | 55 | 188 | 22.212 | fore reef | bioclastic sand | Low |
| Misali Island20 | 65 | 254 | 29.480 | fore reef | bioclastic sand | Low |
| Ras Nungwi peak12 | 56 | 296 | 14.775 | fringing reef | bioclastic sand | Medium |
| Ras Nungwi peak12-14 | 46 | 116 | 7.635 | fringing reef | bioclastic sand | Medium |
| Ras Nungwi peak20 | 81 | 311 | 67.845 | fringing reef | bioclastic sand | Low |
| Ras Nungwi16 | 92 | 408 | 40.674 | fringing reef | bioclastic sand | Medium |
| Ras Nungwi20 | 37 | 76 | 16.497 | fringing reef | bioclastic sand | Low |
| Mnemba Atoll30 | 33 | 87 | 45.218 | sand flat | bioclastic sand | Medium |
| Ocean Paradise3 | 46 | 231 | 57.750 | back reef | bioclastic sand | High |
| Bawe Island9-30 | 80 | 410 | 102.015 | fringing reef | bioclastic sand | High |
| Bawe Island grob | 64 | 308 | 13.077 | fringing reef | bioclastic sand | High |
| Stone Town12 | 77 | 519 | 176.291 | fringing reef | bioclastic sand | High |
| Stone Town20 | 66 | 361 | 158.542 | fringing reef | bioclastic sand | High |
| Menai Bay1 | 36 | 241 | 21.294 | mangrove | fine-grained sand | High |
| Kizimkazi Beach1 | 24 | 59 | 27.949 | sand flat | fine-grained sand | High |
| Mafia outside21 | 44 | 94 | 20.764 | fore reef | bioclastic sand | Medium |
| Mafia outside20 | 82 | 347 | 96.657 | fore reef | bioclastic sand | Medium |
| Chole Bay 1 (18-21) | 27 | 74 | 3.664 | back reef | bioclastic | Medium |

| | | | | | sand | |
|---|---|---|---|---|---|---|
| **Chole Bay 2 (15-18)** | 77 | 241 | 55.658 | fringing reef | bioclastic sand | **Medium** |
| **Chole Bay 2 (20)** | 72 | 281 | 69.383 | fringing reef | bioclastic sand | **Medium** |
| **Mafia Lodge (0-3)** | 62 | 397 | 65.576 | fringing reef | fine-grained sand | **High** |

Figure 3 - the authors show three panels that feature different approaches but the resulting patterns are not dramatically different. Maybe they could just focus on one parameter for the figure and present the caparison (which is still valid to do) in the supplements? Then it could still be discussed in the text but the figure would be easier to read and less crowded.

**Response: We fully understand your concern, however, comparison of the diversity distribution among three order of the Hill numbers is a very important part of our discussion (the entire Section 4.1). Separating Figure 3 into two parts in both the main text and the supplements does not allow direct comparison, on the other hand. The readers may have to repeatedly go through the main text and the supplements at the same time to find the corresponding figures, which may not facilitate their understanding.**

**If the reviewers and editor still strongly prefer the movement of some parts of the display items into the supplements, we are happy to do so later.**

Figure 4 and figure 5 are somehow redundant and have a very similar take home massage if I am correct? Maybe just feature one and put the other in SI?

**Response: We moved the original Figure 4 into the supplements.**

Figure 6-8: I don't understand the figure caption here, but maybe this is just my ignorance...Each figure caption states 'Scanning electron miscoscopy images of the top 10 ostracod species of highest % mean relative abundance for Biofacies 1-4 based on Horn dissimilarity.' But then you get more than 10 species and different species in each figure... please clarify this for me and maybe others that are not experts in the field.

**Response: The original Figures 6-8 together showed the top 10 species of highest % abundance in each biofacies, as listed in Table 2. To avoid misunderstanding and increase clarity, we combined Figures 6-8 into one large figure spanning three pages.**

Minor comments:

Line 27: what is the difference between reefal and algal here? Are algae growing on a reef not considered reefal? Or do you just include calcifying organisms in reefal?

**Response: This is a very good point. There are certainly algae growing on coral reefs, but their percentage of coverage is much lower than that in sand flats and other marginal reef ecosystems. Please see our revised Table 1 for the classification of vegetation coverage of**

each location. By "reefal taxa", we meant any ostracods that predominantly live on mature and healthy coral reefs, while "algal taxa" predominantly inhabit non-reef environments with high algae coverage. Our focus was to differentiate two ostracod faunas between distinct ecosystems instead of between specific plant or animal microhabitats. We revised the text as follows:

**"Highest diversity was found on shallow fringing reefs where coral-affined and algae-affined taxa exhibited maximum overlap of their distributional ranges."**

Line 45-49: While the manuscript is generally well written this sentence is too long and convoluted. Please rephrase.

**Response: We rephrased the sentence to be:**

**"It is considered a useful model organism in modern and paleo biodiversity research because of its high fossilization potential, high abundance, and ubiquity in almost all marine ecosystems (Yasuhara et al., 2017). However, ostracods on coral reefs are poorly understood. Does ostracod exhibit higher diversity in reefal habitats compared with other soft sediment environments? What are the characteristic ostracod taxa occupying different niches on coral reefs?"**

Comments from the Reviewer#2 Peter Frenzel:

The manuscript under review presents a well-executed study in a field poorly known so far - ostracods in tropical reefs. Considering the high importance of ostracods in geosciences, we urgently need baseline studies on reef associations for understanding past and present diversity and distribution patterns. This need is even more pressing for environmental management tasks because of quickly changing conditions for endangered reefal communities in a time of climate change and other anthropogenic threats. The paper presented contributes to this field in an excellent way.

**Response: We appreciate your positive and constructive comments.**

There are only a few remarks from my side. Should be easy to comment on this.

The authors state covering all major types of benthic habitats within their study. What is not clear, however, if typical microhabitats beside bioclastic sands and gravels are covered, e.g. muds, phytal habitats, invertebrate colonies. Also I missed information if brackish water conditions have been encountered, despite Perissocytheridea is mentioned as indicator of brackish water conditions. The discussion of habitats and micro-habitats sampled is significant for discussing biodiversity values.

**Response: Based on our field investigation, we indeed covered all major types of shallow-marine benthic habitats in Zanzibar Archipelago, including fore reef, back reef, fringing reef, sand flat, and mangrove. However, we did not specifically take live ostracods from different microhabitats. The total assemblage comprised of both live and dead ostracods was recovered from surface sediments, representing the time-averaged accumulation of all individuals from all available microhabitats in each location. For every ostracod taxon within the total assemblage, we inferred its microhabitat, e.g., algae, hard coral, or sand bottom, primarily based on literature.**

In addition, we specified the sediment type and algae coverage of each location in Table 1. In terms of sediments, most sites are characterized by bioclastic sands together with reef rubble, and the remaining sites are fine-grained sands. We did not encounter any muddy deposits in our field campaigns. We classified all sites into three categories according to the percentage of algae coverage: high, medium, and low. We hope that the reviewers and readers find such information increases clarity of our manuscript.

Finally, our sampling sites were not in brackish water conditions as there was no river input in this region. *Perissocytheridea* reached high dominance in the mangrove site, which may be explained by its euryhaline nature to adapt to a wide range of salinity in a very shallow intertidal environment. To aid understanding, we now specifically describe the ecology of this genus as "euryhaline" instead of "brackish". All relevant places in the manuscript have been revised.

What is the sample volume or sample area? Abundance data should be given in a standardized way.

Response: This is a very good point. We now report the abundance per gram sediment in Table 1 to include a measure of sample volume.

Table 1 lists species richness and abundance of ostracods per sample. Abundance refers to which area of the bottom or volume of the sample? (see 2) The number of counted individuals would be helpful to interpret species richness.

Response: 'Abundance' as shown in original Table 1 referred to the number of ostracod specimens that were picked and counted in each sample. To avoid misunderstanding, we changed the name of that column to be "No. individuals", and we added another column "Abundance per gram sediment". We also added the following text to discuss this:

"Exceedingly high abundance was found at Stone Town, while sites at Bawe Island and Mafia outside were also abundant, in contrast to the lowest abundance at Haramu Passage and Chole Bay 1 (Table 1)."

In addition, the effect of sample size on diversity measurement was taken into full consideration. By the sampling theory (Chao et al., 2014), we first determined that the largest possible sample completeness across all samples of different sizes is 82.5%, i.e., our observed number of species accounts for 82.5% of true species richness. We then applied rarefaction and extrapolation models based on Hill numbers to standardize samples in terms of sample completeness, which allowed us to make a fair comparison of biodiversity data among samples.

4) Fig. 2 shows a legend with sites attributed to colours. This is probably not needed because sites are also given in (B).

Response: We deleted the color legend of panel (B) accordingly.

Multivariate analysis are highly sophisticated and contribute significantly to interpretations. I missed, however, statements about pretreatment of data. Did you exclude sites from analysis because of low number of specimens or other reasons? Did you exclude taxa from analysis because of very low proportions? Did you exclude highly correlating taxa from analysis?

**Response: We did not exclude any samples from the multivariate analysis because all samples are of relatively high number of ostracods, i.e., more than 50 specimens. We did not exclude any rare taxa from the analysis either. Cutoff of rare species is not considered necessary in our case, since the multivariate analyses performed well with the original census data. Both nMDS and Ward's minimum variance cluster analyses show clear separation of biofacies consistent with our understanding of their ecologies. Furthermore, it is common practice to remove highly correlated environmental variables when testing the correlation, however, scientists do not exclude correlated taxa from the community analysis. Highly correlated taxa likely have similar ecological preferences, and they together as a united ecological group support the delineation of different clusters.**

Fig. 4 and associated discussion: There are two types of habitats covered only one time by the study: sand flats and mangroves. This limits interpretation of these habitats a little. Would be good to addressing this issue with one or two sentences.

**Response: Our sample sites include only one mangrove habitat indeed, but we cover two sand flat habitats in Mnemba Atoll and Kizimkazi Beach, respectively. We added the following text under section 3.1 to address this problem:**

**"Note that the mangrove habitat may be underrepresented in current study as we have only one such site, however."**

Line 299: Write B. elongate

**Response: We revised accordingly.**

Plates with ostracods: Please, indicate samples for figured specimens. Perissocytheridea sp. 1 is probably P. estuaria Benson & Maddocks, 1964.

**Response: Thank you for suggesting the right species name for *Perissocytheridea* species. We have added sample information of each specimen as requested.**

Line 327; Write Bosasella elongate

**Response: We revised accordingly.**

Fig. 8/7 looks quite small compared to fig. 8/8. Are you sure?

**Response: Adult specimens of *Patrizia nucleuspersici* in our samples indeed show some degree of size variation. After careful observation and comparison of all specimens, we tend to think that the larger and smaller individuals all belong to the same species. We**

**understand your point at the same time, so we replace Fig. 8/7 with another specimen of normal size.**

Lines 424-425: Make clear distribution is based on your study and not necessarily reflecting all habitat types and regions of the study area.

**Response: We rephrased the sentence to be:**

**"Thus, our study indicates that the distribution of shallow-marine ostracods in Zanzibar Archipelago is characterized by three reefal facies and one intertidal facies."**

Line 472: add space in Figs. 9-10

**Response: We added space as suggested.**

Lines 493-498: Maybe because ostracods are using different microhabitats and food sources than forams?

**Response: This is a very valuable point. Both ostracods and foraminifera in shallow-marine environments can be broadly classified into two major types, which are phytal and sediment-dwelling taxa (Fajemila et al., 2020; Kamiya, 1988; Langer et al., 1989). More detailed knowledge about the microhabitat of individual species is, however, not available in most cases. We think that the usage of food resources may be more important in explaining the differences in biodiversity and biogeographic patterns between these two groups. We added the following text to discuss this:**

**"Another important factor accounting for the different distributional patterns between ostracods and foraminifera is likely their tolerance to eutrophication and pollution. Previous studies indicate that an intermediate level of eutrophication is beneficial to ostracods and many other soft sediment benthos, which are also resistant to heavy metal pollution in highly urbanized areas (Hong et al., 2022). Consistently, our sampling sites at Stone Town reported the highest abundance of ostracods (Table 1). Foraminifera on the other hand are susceptible to environmental stressors, as shown by low taxonomic richness and high dominance of the faunas in eutrophic conditions (Mamo et al., 2023). In our case, it makes sense that the highest diversity of foraminifera was found in pristine and oligotrophic Pemba waters."**

14) Line 554: Which experiments are mentioned here?

**Response: By "experiment", we meant all the lab work to process the sediment, and pick and identify the ostracods. We revised the text to be:**

**"SYT performed the research and collected the data."**

15) References: There are quite a number of spelling mistakes in the references Hartmann 1974, Hsieh et al. 2016, Jellinek 1993

**Response: We fixed the spelling mistakes.**

Comments from the Reviewer#3 Ilaria Mazzini:

The authors contribute significant insights into a relatively neglected aspect of the reef ecosystem with their investigation titled 'Reefal ostracod assemblages from the Zanzibar Archipelago (Tanzania)'. Their research establishes a foundational understanding, albeit one influenced by human activity, of this functional group's presence along the eastern Tanzanian coastline. The manuscript contributes to assess the diversity of the endangered reefal assemblages, providing valuable information.

**Response: We really appreciate your positive comments.**

Since sampling with a plastic bag underwater implies the loss of the finer particles due to suspension, at deep sites such as Mapenduzi wall and the Haramu passage for instance (below 30m of depth), I think that a better characterization of the sampling sites would be important. More detailed description of the sampling/processing would also be helpful (standard volume/ weight?).

**Response: We were fully aware of the problem of suspension during our sample collection. We just did not explicitly describe our sampling methods in the original manuscript. As suggested, we rephrase the relevant text to be:**

**"Samples were collected by SCUBA diving to scrape along the seabed and fill plastic containers with surface sediments from the top 2 cm, in order to avoid the loss of finer particles due to suspension."**

**We agree that a better characterization of each sampling site is important. We thus add three additional columns in Table 1 to list "Abundance per gram sediment", "Sediment", and "Algae coverage" of each sample. We hope you find that such information helps understanding.**

The authors state that "Pemba reefs are likely in pristine conditions with the highest coverage of live hard corals, while Zanzibar reefs are often dominated by dead corals intermingled with algae and seagrass habitats". I think a better characterization of the sampling sites would be very helpful also for comparisons in future studies.

**Response: Please refer to our reply to the above comment. We characterize the algae coverage of each site in Table 1.**

There is one mangrove site (Menai Bay) and one sand flat site (Kizimkazi Beach) and they are characterized by biofacies 3 as Ocean Paradise, a back reef site whereas Chole Bay 1, another back reef site, is characterized by biofacies 4. This is not discussed, at all.

**Response: The back reef of Chole Bay 1 is grouped with typical fringing reefs in Biofacies 2. They all belong to the same type of habitat in terms of intermediate water depths and relatively high algae coverage. Therefore, the assignment of Chole Bay 1 as a back reef into Biofacies 2 is consistent with our understanding of the ecological situation here. On the other hand, Biofacies 3 is characterized with very shallow non-reef and marginal-reef**

habitats in the intertidal zone. The back reef site, Ocean Paradise, is from 3 m water depth, so it fits the main ecological features of Biofacies 3. The separation of two back reef sites into two different clusters indicates the importance of water depth overwhelming that of habitat type in this case. As the diversity and composition of benthic faunas are influenced by a set of environmental factors at the same time, we do not expect the delineation of clusters would strictly reflect habitat type. For example, Biofacies 1 includes mostly fore reefs but also a fringing reef site (Kokota Reef). However, we do not think it is practical to discuss such very minor inconsistencies in every detail, as the manuscript would be overcrowded and distractive in this way.

At the same time, we fully understand your point. We revised the discussion about the habitats of Biofacies 2 and 3 for better clarity and coherence:

"Biofacies 2 mostly represented a different type of relatively shallow (12-20 m) fringing-back reef habitats of Ras Nungwi, Chole Bay, and Mafia outside, in addition to a deeper (30 m) sand flat of Mnemba Atoll."

"The remaining Biofacies 3 corresponded to the shallowest intertidal habitats with various benthic communities, including a marginal back reef, a marginal fringing reef, a sand flat, and a mangrove."

I am impressed by the statistical analysis. But I think the presentation of the statistical analyses could be better organized. For instance, Fig. 3 is never cited in the text…then is it necessary or could it be included in the supplementary material? Or figs 3 and 5 could be merged figuring only the most significant D and dissimilarity index.

Response: We agree with the reviewer that our display items could be better presented and organized. However, we indeed cited Fig. 3 at Line 217 in the first place, in addition to many other places. We hesitate to delete any panels from Figure 3 and 5 in order to combine them. It is because the comparison of the diversity and cluster distribution among all three orders of Hill numbers is a very important part of our discussion. We spent the entire Section 4.1 and the first paragraph under Section 4.2 focusing on this. Separating Figure 3 into two parts in both the main text and the supplements does not allow direct comparison of the patterns across different order q. The readers may have to repeatedly go through the main text and the supplements at the same time to find the corresponding figures. For these reasons, we tend to keep Figure 3 and 5 as they are. We specifically cite every figure at all relevant places throughout the manuscript.

The plates are really nice (Figs 6-7-8) but there is no reference to them in the text. Add a reference or put them in supplementary. Specify to which sample belongs each figured specimen.

Response: We referred to Figs 6-8 at Line 297 in the first place. After reordering, all SEM pictures are in the new Figure 5 now. We added the origin of each specimen from which sample as requested.

I have few remarks, listed below:

Line 124

Please indicate during when (month and year) the sampling took place (to be added in table 1).

**Response: We added this information into supplementary table 1 as requested.**

Line 135

Small species can form a large part of the assemblage. See for instance Aiello et al. 2024 https://doi.org/10.1016/j.revmic.2023.100755

**Response: Thank you for recommending this excellent paper. By dry sieving through a 150 μm mesh size, we might miss the early stage juveniles of smaller species. We will keep this in mind and consider using different sample processing methods in the future. In the case of the current study, we tend to think that picking from >150 μm fraction is acceptable since it is still the common practice nowadays.**

I agree that you studied time-averaged assemblages but at least an indication on the living assemblage (how many specimens with soft parts in each sample) would be helpful. Moreover, what is the average volume of sediment you processed?

**Response: We did not dye the sediment after collection, so it is impossible to measure the exact ratio of living ostracods at this stage. The percentage of specimens with soft parts is very low in every sample. We added further information regarding this:**

**"The sample materials were primarily death assemblages though a very small number of specimens were preserved with soft parts (less than 1% among all observed individuals), indicating they were alive at the time of collection."**

**We now report the abundance per gram sediment in Table 1 to include a measure of sample volume.**

Line 136

"sediment rich samples" means? Which kind of sediment? Coral sand, coral gravel and dead coral fragments (devoid of algal covering, with algal covering?). Were the samples different in this aspect?

A better characterization of the sampling sites would also help understanding the discussion and the gradient of diversity of the ostracod assemblage from coral reefs to algae turfs.

**Response: "sediment rich" simply meant that we collected a large amount of sediment (e.g., 500 gram) from a site and it is not practical to check all sediments for ostracods. It has nothing to do with sediment type, size fraction, or any other parameters. We rephrased the sentence to be:**

**"Large volume samples were split into aliquot fractions using a microsplitter."**

**Please see Table 1 for the characterization of sample sites.**

Reference list: there are many spelling errors, in the german literature especially, please check

them.

**Response: We fixed the spelling mistakes.**

**End of letter**

**Sincerely yours**

**Corresponding author Skye Yunshu Tian, Martin Langer**

**References**

Chao, A., Gotelli, N. J., Hsieh, T., Sander, E. L., Ma, K., Colwell, R. K., & Ellison, A. M. (2014). Rarefaction and extrapolation with Hill numbers: a framework for sampling and estimation in species diversity studies. *Ecological Monographs*, *84*(1), 45-67. https://doi.org/https://doi.org/10.1890/13-0133.1

Fajemila, O. T., Sariaslan, N., & Langer, M. R. (2020). Spatial distribution of benthic foraminifera in the Lagos Lagoon (Nigeria): Tracing the impact of environmental perturbations. *PLoS One*, *15*(12), e0243481.

Kamiya, T. (1988). Morphological and ethological adaptations of Ostracoda to microhabitats in Zostera beds. *Developments in Palaeontology and Stratigraphy*, *11*, 303-318. https://doi.org/https://doi.org/10.1016/S0920-5446(08)70191-2

Langer, M., Hottinger, L., & Huber, B. (1989). Functional morphology in low-diverse benthic foraminiferal assemblages from tidal flats of the North Sea. *Senckenbergiana maritima*, *20*(3-4), 81-99.